# Monosex Populations of the Giant Freshwater Prawn *Macrobrachium rosenbergii*—From a Pre-Molecular Start to the Next Generation Era

**DOI:** 10.3390/ijms242417433

**Published:** 2023-12-13

**Authors:** Melody Wahl, Tom Levy, Tomer Ventura, Amir Sagi

**Affiliations:** 1Department of Life Sciences, Ben-Gurion University of the Negev, P.O. Box 653, Beer-Sheva 8410501, Israel; wahlm@post.bgu.ac.il; 2Institute for Stem Cell Biology and Regenerative Medicine, School of Medicine, Stanford University, Stanford, CA 94305, USA; levyt@stanford.edu; 3Hopkins Marine Station, Stanford University, Pacific Grove, CA 93950, USA; 4Centre for Bioinnovation, University of the Sunshine Coast, Sippy Downs, QLD 4556, Australia; tventura@usc.edu.au; 5School of Science and Engineering, University of the Sunshine Coast, Sippy Downs, QLD 4556, Australia; 6National Institute for Biotechnology in the Negev, Ben-Gurion University of the Negev, P.O. Box 653, Beer-Sheva 8410501, Israel

**Keywords:** androgenic gland, decapods, environmental management, IAG-switch, insulin-like peptide, sexual manipulation

## Abstract

Sexual manipulation in the giant freshwater prawn *Macrobrachium rosenbergii* has proven successful in generating monosex (both all-male and all-female) populations for aquaculture using a crustacean-specific endocrine gland, the androgenic gland (AG), which serves as a key masculinizing factor by producing and secreting an insulin-like AG hormone (IAG). Here, we provide a summary of the advancements from the discovery of the AG and IAG in decapods through to the development of monosex populations in *M. rosenbergii*. We discuss the broader sexual development pathway, which is highly divergent across decapods, and provide our future perspective on the utility of novel genetic and genomic tools in promoting refined approaches towards monosex biotechnology. Finally, the future potential benefits of deploying monosex prawn populations for environmental management are discussed.

## 1. The AG and IAG-Switch in Crustaceans

A brief history of the discoveries of the AG and the IAG in crustaceans, with an emphasis on decapods.

In vertebrates, sexual development cascades are well characterized, including the pivotal role of sex steroids. Studies have investigated the presence of sex steroids and their potential role in sexual development in crustaceans. Sex steroids, including androgen, estrogen, and progestogen, have been identified in decapod tissues, such as the gonads and eyestalk [1,2,3,4]. Although some studies have explored the effects of hormone treatments on sexual development and reproductive processes [5,6,7,8], the literature regarding the underlying mechanisms and factors influencing the outcomes remains limited and lacks clear evidence.

Instead, an endocrine gland unique to male crustaceans from the class Malacostraca was shown to establish masculinization. The first mention of the gland was in 1947 in the male reproductive system of the blue swimming crab *Callinectes sapidus Rathbun* [9], followed by its discovery in many other decapod crustaceans (Figure 1A,B), while its function was deduced by the early work of [10] in the amphipod *Orchestia gammarella*, demonstrating for the first time an androgenic function of the unique gland, thus named the androgenic gland (AG) [10]. Bilateral removal of the AGs from *O. gammarella* males ceased the differentiation of secondary characteristics and caused a decrease in spermatogenesis. In the reciprocal study, AG implantation into *O. gammarella* females caused masculinization of the primary and secondary sexual characteristics. However, gonads implanted into the opposite sex had no effect on the host. These studies led to the conclusion that the AG is the exclusive source of a hormone responsible for the development of primary and secondary male characteristics [10,11,12]. In decapods, similar results validated the conserved key role of the AG in masculinization; immature and mature giant freshwater prawn (*Macrobrachium rosenbergii*) females implanted with AG developed appendices masculinae (male-specific protrusions that develop bilaterally on the second pair of swimming legs), while females implanted with vas deferens or testicular tissue developed normally [13]. In the red swamp crayfish, *Procambarus clarkii,* AG implantation into immature females inhibited vitellogenesis [14], while ovarian regression ensued in AG implanted females of the mud crab *Scylla paramamosain* [15]. In the Chinese mitten crab, *Eriocheir sinensis*, AG extracts from *S. paramamosain* and *E. sinensis* injected into females caused the development of external male characteristics [16].

The attempts to isolate the AG hormone raised a debate with regard to the chemical nature of the active substance, with evidence suggesting either a protein or a lipidic nature. In isopods, Katakura (1975) partially purified the active protein from the AG of *Armadillidium vulgare* and showed that a single injection of the extract into females induced masculinization of the secondary sexual characteristics [17]. The lipidic and steroidal extracts of the AG did not induce masculinization [18]. A series of studies in the isopod, *A. vulgare*, purified and elucidated the structure of an androgenic hormone [19,20]. Isolation and characterization of a cDNA encoding a precursor of the hormone showed a resemblance to the insulin superfamily structure, comprised of a signal peptide, B chain, C peptide, and A chain [21]. Later, similar androgenic hormones with highly conserved protein sequences were identified in two additional isopod species, *Porcellio scaber* and *Porcellio dilatatus* [22]. In decapods, biochemical studies [23] and histological observations [24] of lipidic substances in the AG suggested that steroids act as androgenic hormones. Efforts to find genes orthologous to the isopods’ androgenic hormones in the group of decapods have failed. It was not until 2007 when the first AG-specific gene, expressed exclusively in males, was discovered in the crayfish *Cherax quadricarinatus*, termed insulin-like AG factor (IAG) [25] (Figure 1C), followed by its discovery in *M. rosenbergii* [26], demonstrating structural similarity to the insulin superfamily (Appendix A). The identification of the first decapod IAG-encoding gene signifies a longstanding gap in our understanding of the AG and its function in decapods, following decades since the discovery of an androgenic hormone in isopods. The finding was enabled through a subtractive cDNA library prior to the prevalent use of next-generation sequencing. Since the discovery of IAG in these two species, IAG-encoding sequences have been identified in a wide range of decapod species including gonochoristic, hermaphrodites [27,28], and parthenogenic [29], of commercial and ecological significance, greatly facilitated by the next-generation sequencing, which is commonly used nowadays.

## 2. Manipulating the IAG-Switch

From early surgical attempts at the AG level up to state-of-the-art molecular manipulations to achieve full and functional sex reversal

Early studies of AG ablation and implantation established the crustacean AG as a key sexual differentiating organ. The IAG hormone secreted from the AG serves as a master switch in decapod crustaceans, in which IAG expression induces masculinization and its absence results in feminization, thus termed the ”IAG-switch” [30].

Moreover, unlike vertebrates, in male crustaceans, the endocrine and gametogenic functions are separated into two distinct organs, the AG and the testis, respectively [31,32]; thus, manipulation of the AG can be performed without affecting the gonad.

With respect to sexual plasticity, one of the most studied crustaceans is *M. rosenbergii*, which displays a fascinating social hierarchy [33], with distinctive morphotypic differentiation of the males and clear differences between males and females in growth patterns (Figure 2A,B). Since males grow faster than females, an interest in developing all-male populations was explored based on a small-scale cage experiment showing that all-male populations might provide double the yield compared with mixed and all-female populations [34] (Figure 2C). The above triggered experiments of bilateral AG ablation at an early juvenile developmental stage, which resulted in complete sex reversal in functional neo-females. When mated with normal males, these neo-females produced all-male progeny [35]. The timing of AG removal was shown to be critical for the manipulation to succeed. *M. rosenbergii* males that were andrectomized in the youngest developmental stage exhibited complete feminization, including the development of oviducts and female gonopores and the initiation of oogenesis, in contrast to later developmental stages in which males were either partially feminized or not feminized at all [36]. The first mass production of all-male populations was established by microsurgical AG ablation at early differentiation stages; however, the production process had drawbacks such as low microsurgery success rates, long duration between microsurgery, and identification of neo-females and intensive labor demand [37]. It became evident, though, that a biotechnology based on manipulation of this endocrine gland could provide the long-sought solution for monosex population aquaculture.

The needed transformation from surgical manipulations to molecular methods was completed through the use of RNA interference (RNAi) gene knockdown [38]. This method has contributed to clarifying the functionality of many crustacean genes of importance to metabolism, development, growth, and reproduction [39]. The RNAi-based biotechnology was a game-changer in the production of *M. rosenbergii* monosex populations. Initial use of this method to silence the *IAG* in mature *M. rosenbergii* males showed reduced spermatogenesis and caused AG hypertrophy [26]. Given the temporal effect and non-GMO nature of the intervention [40], manipulating the expression of the *IAG*-switch was suggested as a replacement for the microsurgical removal of the AG by using *IAG* silencing prior to sexual differentiation.

*M. rosenbergii*, like many Malacostraca species, bears the WZ/ZZ system of sexual heritability, in which females are heterogametic (WZ) and males are homogametic (ZZ) [41,42,43,44,45]. A key component in detecting successful IAG-switch manipulations was the use of genetic sex markers. These were not available for *M. rosenbergii* until 2010 [46], when the use of the cumbersome amplified fragment length polymorphism (AFLP) enabled the identification of the first genetic sex markers for this species. With the advent of restriction-assisted DNA sequencing (RAD-Seq) and RNA-Seq, genetic sex marker identification is streamlined and has indeed contributed to the discovery of these markers in multiple decapod species, either within the WZ/ZZ or XX/XY systems [30,47]. Using genetic sex markers, successful sex reversal of genetic ZZ males into functional ZZ neo-females was first achieved through gene silencing in 2012 [48]. This technology now enables large-scale production of all-male populations using RNAi, the first application of this method in the entire field of aquaculture.

*M. rosenbergii* presents clear differences between sexes. Males display morphotypic differentiation while females grow more uniformly, making monosex a clear advantage. The AG is known to be key for this change in behavior, as was exemplified in *C. quadricarinatus*, where AG implanted into females induces male-like behaviors [49]. While sex-specific behaviors remain largely unexplored across farmed decapods, evidence suggests monosex would be advantageous across decapods, as, for example, in the case of the commercially most important marine shrimp, *Litopenaeus vannamei*, where subtle sex-biased behaviors were recorded [50].

Although in *M. rosenbergii*, some of the males grow larger than females and reach the highest sizes in a mixed population, thus providing economic benefits [51,52,53], all-female culture was suggested to be more favorable under intensified conditions [54] (Figure 2D). Production of all-male populations requires stocking in low densities due to aggressiveness and territoriality. Males display a high variation in size distribution with strong dominance in hierarchical social structure [33,55]. While females do not reach the size of the dominant males, they grow more uniformly, as initially proposed in 1992 [43]. Manipulation of the AG using surgical implantation of AG tissue from adult males into early developmental stage females resulted in a fully functional reversal of sex to neo-males. However, the survival and success rates of the functional neo-males were only ∼10% [43]. Malecha (2012) claimed that the efficiency of producing all-female populations depends on increasing the success rate of producing WZ neo-males via the use of exogenous AG materials [54]. In 2016, through molecular manipulation of the IAG-switch, Levy et al. (2016) demonstrated a fully functional sex reversal of WZ females into WZ neo-males by a single injection of suspended hypertrophied AG cells [42]. Verification of the successful generation of neo-males was enabled using genetic sex markers [46]. Above 80% of cell-injected females had developed both *appendix masculina* and male gonopores. The injected sex-reversed females functioned as males and were reproductive, thus considered neo-males [42]. This successful all-female biotechnology has one drawback of being dependent on the relatively small amount of AG cells found in a donor body; thus, lentiviral-transduced ectopic expression of IAG in non-AG primary cell culture was attempted to increase the volume of relevant cells for the process and avoid the male donor dependency in this biotechnology [56]. This is a preliminary step to be further tested for applicability.

## 3. Controlling Elements of the IAG-Switch

Since the discovery of the first decapod IAG [25], it has been found in a variety of crustacean species, including prawns, shrimp, crayfish, lobsters, and crabs [30], providing a wide context within crustaceans and a starting point for the challenge of discovering the bridging cascade upstream of the IAG-switch. To date, IAG has enabled full sex change only in *M. rosenbergii*, suggesting this molecular switch needs to be further explored to fulfill its potential in other species. The path towards such discoveries of mechanisms upstream of the IAG-switch is now open and could be studied using functional genomics tools such as RNAi and CRISPR. Another interesting species with the above respects is the crayfish *C.* quadricarinatus, in which sexual plasticity is reflected in naturally occurring WZ-genotyped intersexual animals with an active male reproductive system and male secondary sexual characteristics, along with an inactive ovary (i.e., naturally born WZ neo-males). Intersexuality was manipulated by IAG silencing, which resulted in a sexual shift. This IAG-switch manipulation resulted in male feature feminization, vitellogenin expression, and oocytes with yolk accumulation [57]. Moreover, findings in hermaphrodites, particularly in protandric species, suggest the IAG-switch plays a critical role in the natural sex reversal path. Unlike gonochoristic species, the sexual differentiation process in hermaphrodites does not relate to the early developmental stages but happens at the mature life stage with a sexual transformation from maleness to femaleness. A study of the northern spot shrimp, *Pandalus platyceros*, describes the IAG-switch using four stages: juveniles, adult males, transitionals, and in adult females [28]. The *IAG* temporal expression pattern in the AG was the highest in juveniles, declined in adult males, and was found to be negligible during the transitional phase and adult females. This suggests that the IAG has a crucial role in the early male differentiation and maturation stage. Moreover, *IAG* loss of function through RNAi in mature *P.* platyceros males induced the masculine to feminine sexual transformation that naturally occurs in *this protandric species*, supporting the pivotal role of the IAG-switch in regulating the transformation between adult stages in hermaphrodite species [28]. In *M. rosenbergii* adult stages, IAG was found to be correlated with the reproductive readiness of male morphotypes. The expression levels of *Mr-IAG* in the reproductively less active orange-claw males were significantly lower than in the blue-clawed males and small males, suggesting a key role for IAG in regulating morphotypic differentiation [58].

The research on the upstream controlling elements of sexual differentiation through the IAG-switch is in progress, with functional studies suggesting diverse candidates for this role (Table 1). In mature specimens of gonochoristic species, it seems that the IAG is controlled as part of the eyestalk-AG-testis endocrine axis [59]. Within this axis, *IAG* is thought to be controlled by upstream neuropeptides that are produced in the X-organ (XO) and accumulated in the sinus gland (SG; both the XO and SG reside in the eyestalk, forming a neuroendocrine complex known as the XO-SG), from where they are secreted. The fact that eyestalk ablation in males causes hypertrophy of the AG [59,60] and *IAG* over-expression [61] indicates that eyestalk neuropeptides serve as upstream controlling elements of AG activity in adults [59,62]. Recent studies in *E. sinensis* [63] and *S. paramamosain* [64] suggest regulatory feedback between the Crustacean Female Sex Hormone (*CFSH*) in the eyestalk and *IAG*. In *EsCFSH-1*-silenced *E. sinensis* mature males, the expression of *EsIAG* in the AG was significantly increased, while *EsIAG* knockdown significantly increased the expression of *EsCFSH-1* in the male eyestalk [63]. In *S. paramamosain*, in vitro treatment with recombinant SpCFSH protein in AG significantly decreased the mRNA levels of the signal transducers and activators of transcription (STAT)-binding site in the IAG promoter. Based on these results, it was suggested that CFSH acts as an inhibiting factor by suppressing the expression of *SpSTAT*, which then regulates *SpIAG* expression [64]. In the protandric hermaphrodite peppermint shrimp, *Lysmata vittata*, injection of recombinant CFSH1b suppressed *IAG* expression, and *Lvit-CFSH* knockdown stimulated the male phenotype development [65]. Other eyestalk-borne neuropeptides suggested to play a role in the eyestalk-AG-testis axis are members of the crustacean hyperglycemic hormone (CHH) superfamily of neuropeptides. In the Pacific white leg shrimp *L. vannamei*, RNAi of two *CHH* genes, *LvCHH1* or *LvCHH2* induced elevated *LvIAG* expression, while injection of their recombinant protein reduced *LvIAG* expression, suggesting inhibitory regulation of CHHs over *IAG* [66].

Sexual development is a succession of three processes, starting with sex determination occurring at the establishment of the zygote, which leads to the sexual differentiation process, followed by sexual maturation (Figure 3). While the AG is regarded as a sexual differentiating organ, the sex-determining mechanism is considered the primary process that regulates the development and function of the AG [67,68]. Several candidates were suggested as the transmitters between sex determination and sexual differentiation in decapods. In *C. quadricarinatus*, Sxl (sex-lethal) alternatively-spliced variants were suggested to be involved in the mechanism of male sex determination/differentiation, with a gradually increased expression pattern in embryonic stages, higher transcript levels at early-stage testis development, and significantly reduced *CqIAG* expression levels following *CqSxl3* silencing [69]. In the mud crab, *S. paramamosain,* the transcription factor Doublesex (*Dsx*) promotes *Sp-IAG* expression, while the transcription factor forkhead L2 (*foxl-2*) inhibits it, as demonstrated in both *in vitro* cell experiments and *in vivo* RNAi studies [70]. Knockdown of the invertebrate double-sex and mab-3 related transcription factor 2 (*idmrt-2*) in *S. paramamosain* decreased the expression of *Dmrt-like* and *foxl-2* genes in the testis, and *IAG* in the AG [71]. In the black tiger shrimp, *Penaeus monodon*, RNAi of *Dsx* significantly decreased the expression of *PmIAG* in the testis and was suggested as a positive regulator of *IAG* by specific binding upstream of the *IAG* promoter [72]. Similarly, *Dsx* was suggested to regulate *IAG* in the shrimp *Fenneropenaeus chinensis*, as knockdown of *FcDsx* resulted in *FcIAG* transcript down-regulation [73]. In *E. sinensis*, RNAi of Es*Dsx1*, Esi*DMY*, and Es*iDmrt1a* reduced *IAG* transcription levels [74]. Direct interaction of the EsDsx-like protein with the *EsIAG* promoter suggests its role as an upstream regulator of *IAG* [75]. Contrary to this, knockdown of *C. quadricarinatus Dsx* induced *CqIAG* transcript levels [76]. In *M. rosenbergii*, *MrDsx* expression was significantly induced in the testis and AG following eyestalk ablation, and gene knockdown of *MrDsx* resulted in a significant decrease in *MrIAG* transcript levels in the AG, suggesting that eyestalk elements negatively control *MrDsx*, which regulates the activation of *MrIAG* [77]. As part of the extensive research aimed at elucidating the IAG signaling cascade in decapod species, IAG loss-of-function studies have played a significant role in the understanding of the IAG downstream factors [78,79,80,81,82,83]. Still, in most of the studies on candidates upstream of the IAG-switch, it seems that the intervention (usually RNAi) is too late to effectively influence sexual differentiation. That can be potentially resolved by an oocyte-specific delivery (OSDel) tool for gene silencing at the oocyte stage prior to egg laying [84]. Yet, the barrier of sex identification at an early stage might be the main obstacle to understanding the sexual differentiation mechanism in other decapod species that might be even more variable.

**Table 1 ijms-24-17433-t001:** Genes suggested as sexual development regulation upstream of the IAG-Switch.

Gene	Species	Tested Tissue or Stage	Sex Heritability Mechanism	Effect on Sexual Differentiation Processes	Reference
*Dmrt11E*	*Macrobrachium rosenbergii*	Post larvae	WZ/ZZ	*MroDmrt11E* knockdown induced a functional sex reversal	[85]
		Terminal ampullae	WZ/ZZ	*MroDmrt11E* knockdown reduced *MrIAG* expression	[86]
	*Macrobrachium nipponense*	Ovary and hepatopancreas (females) Abdominal ganglia (males)	WZ/ZZ	*MniDMRT11E* knockdown reduced *MnVG* expression in females and increased *MnIAG* expression in males	[87]
*Dsx*	*Fenneropenaeus chinensis*	Male cephalothorax	WZ/ZZ	Two isoforms of *FcIAG* gene were down-regulated following *FcDsx* knockdown	[73]
	*Macrobrachium rosenbergii*	AG	WZ/ZZ	*MrDsx* knockdown reduced *MrIAG* expression	[77]
	*Cherax quadricarinatus*	Cephalothoraxes of undifferentiated crayfish	WZ/ZZ	*CqDsx* knockdown decreased *CqIAG* expression	[76]
* SOXE *	*Portunus* *trituberculatus*	AG and testis	XY/XX	*PtSoxE* siRNA reduced *PtIAG* and *PtIR* expression	[88]
*SOXB2-1*	*Eriocheir sinensis*	Testis	WZ/ZZ	*EsSoxB2-1* knockdown led to abnormal nucleus transformation during spermiogenesis	[89]
*SOX9*	*Scylla paramamosain*	Cell culture	WZ/ZZ	Sox9 binding site mutation in *SpVIH* promoter reduced activity	[90]
* CHH1 *	*Litopenaeus vannamei*	AG	WZ/ZZ	Knockdown of either *LvCHH1* or *LvCHH2* resulted in increased *LvIAG* expression; injection of their recombinant protein led to decreased *LvIAG* expression.	[66]
* CHH2 *					
* GC receptor *	*Litopenaeus vannamei*	AG	WZ/ZZ	*LvGC* knockdown increased *LvIAG* expression	[66]
* MIH *	*Macrobrachium nipponense*	AG	WZ/ZZ	Gene knockdown increased *IAG* expression	[91]
*GIH* (*VIH*)					
*Masc*	*Macrobrachium rosenbergii*	Post larvae	WZ/ZZ	*MrMasc* knockdown obtained full sex reversal; Insulin-like signal pathway has been identified in *MrMasc* knockdown prawns.	[92]
*Sxl*	*Cherax quadricarinatus*	Undifferentiated juvenile prawns	WZ/ZZ	*Sxl3* knockdown decreased *CqIAG* expression	[69]
*Foxl2*	*Scylla paramamosain*	Ovaries	WZ/ZZ	*Spfoxl2* knockdown increased *SpVG* expression	[93]
*CFSH*	*Portunus* *trituberculatus*	Eyestalk, AG and testis	XY/XX	*PtCFSH* knockdown increased *PtIAG* expression; injection of its recombinant protein led to decreased *PtIAG* expression.	[94]
	*Scylla paramamosain*	Cultured AG	WZ/ZZ	Recombinant SpCFSH reduced *SpIAG* expression	[95]
		Cultured AG	WZ/ZZ	Recombinant SpCFSH reduced *SpSTAT* expression	[64]
* STAT *	*Scylla paramamosain*	Cultured AG	WZ/ZZ	*SpSTAT* knockdown reduced *SpIAG* expression	[64]
*Tra2*	*Macrobrachium nipponense*	Gonads	WZ/ZZ	*MnTra2* Inhibited *MnSxl* expression	[96]
* GEM *	*Macrobrachium nipponense*	AG and testis	WZ/ZZ	*MnGEM* knockdown increased *MnIAG* expression and testosterone content	[97]
* BMP receptor *	*Scylla paramamosain*	Pre/early/late vitellogenic females	WZ/ZZ	*SpBMPR* knockdown decreased vitellogenin receptor expression	[98]

Abbreviations: AG, androgenic gland; BMP, bone morphogenetic protein; CFSH, crustacean female sex hormone; CHH, crustacean hyperglycemic hormone; Dmrt11E, doublesex and mab-3 related transcription factor 11E; Dsx, doublesex; Foxl2, forkhead box protein L2; GC, Guanylate cyclase; GEM, gem-associated protein 2-like isoform X1; GIH, gonad inhibiting hormone; IAG, insulin-like androgenic gland hormone; IR, insulin-like receptor; Masc, masculinizer; MIH, molt-inhibiting hormone; SOX9, SRY-Box Transcription Factor 9; SOXB2-1, SRY-related HMG box group B2; SOXE SRY-related HMG box group E; STAT, signal transducers and activators of transcription; Sxl, sex-lethal; Tra2, transformer-2; VG, vitellogenin; VIH, vitellogenin inhibiting hormone.

## 4. Distinct Biotechnologies for Monosex Populations of Prawns

Description of the two biotechnologies and their respective uses at different husbandry options.

Through manipulations of the IAG-switch, a high degree of induced sexual plasticity in *M. rosenbergii* has been demonstrated. The manipulations afforded cases of functional WZ males, ZZ females, and, surprisingly, even WW males and females [42,48,99]. IAG-switch manipulations led to the pioneering development of two biotechnologies for *M. rosenbergii*: all-male [48] and all-female [42] aquaculture. The RNAi-based biotechnology for all-male aquaculture through *dsIAG* injection [100] facilitated a large-scale production of all-male *M. rosenbergii* with 86% success sex-reversal of ZZ male to ZZ neo-female, then mating the neo-females with normal males, yielding 100% of all ZZ male progeny exhibiting typical population structure of *M. rosenbergii* male morphotype differentiation [33,40]. Together with the population structure, the temporal effect of the dsRNA in the target tissue proved the safety of RNAi usage in all-male production [40], the first commercial use of RNAi in the entire field of aquaculture. Shpak et al. showed the application of the technology beyond one all-male generation and established three successive generations of W chromosome-free ZZ prawn lines [101]. Up to date, over 15 consecutive W-free generations have been reported in this population (Wahl, personal communication). The third generation of all-males was comparable to males from a normal mixed population. The typical morphology of the testicular lobes, spermatophore, and AG was demonstrated by a histology of the testis and terminal ampulla in males from both populations. Gene expression patterns of male-specific genes such as *Mr-IAG* and *Mr-Mrr* were similar in males from the all-male third generations compared to males from a normal mixed population [101].

Correspondingly, the first biotechnology for all-female aquaculture by a single injection of hypertrophied AG cell suspension enabled, for the first time, all-female mass production in three steps: (1) WZ females were sex reversed into WZ neo-males by injection of suspended hypertrophied AG cells. (2) crossing of WZ neo-males with normal WZ females yielding a progeny with 25% WW females; (3) crossing the WW females with normal ZZ males yielding a 100% WZ female population [42]. This technology enabled, for the first time, a comparative large-scale field study of all-female versus mixed populations under extensive and intensive stocking conditions without the need for manual segregation. This field study, in which all-female progenies were cultured without males from Day 0, showed that all-female culture had superior performance to that of mixed culture with respect to feed conversion ratios, survival rates, total yield, and uniform body size, with stocking densities up to 4 times those usually practiced with mixed sex and all-male populations of the species [102]. However, the road to a viable breeding stock for all-female *M.* rosenbergii production necessitated a more efficient way to produce WW females via further manipulations of the IAG-switch to achieve WW neo-males. Indeed, this final manipulation of the sexual switch yielded WW females that were sex-reversed by injection of suspended hypertrophied AG cells into fully functional WW neo-males, which were then crossed with WW females to produce a Z chromosome-free breeding stock population [99]. The case of all-female consecutive generations without the Z chromosome was further investigated for three generations, showing that the performances of the WW all-female population were comparable to the WZ all-female population in survival rate, size uniformity, body weight, and yield [103]. This Z-free population has been managed with no visible abnormalities for over 8 generations (Wahl, personal communication).

## 5. Molecular and Genomic Implications

The availability of NGS technologies yielded novel molecular and genomic tools that were applied to several crustacean species [104]. Ample transcriptomic libraries of decapod crustaceans were studied, with several referring to sexually biased transcripts [27,28,105,106,107,108], including the description of different stages in the life cycle of *M. rosenbergii* [109,110,111]. This is true also at the genomic level, with published genomes of decapod species, including the crayfish *Procambarus virginalis* [112] and *C. quadricarinatus* [113], the crabs *E. sinensis* [114,115], *Birgus latro* and *Paralithodes camtschaticus* [116], the ridgetail white prawn *Exopalaemon carinicauda* [117], the shrimp *Neocaridina denticulata* [118], *L. vannamei* [119], *Marsupenaeus japonicus* and *Penaeus monodon* [120], and the lobsters *Panulirus ornatus* [116] and *Homarus americanus* [121]. A high-quality *M.* rosenbergii genome, exhibiting distinguishable paternal and maternal sequences and enabling the identification of W/Z-specific sequences [99], was also studied for future mapping at the chromosome level with resolved chromatids and SNP mapping.

A major reliance of molecular functional research is based on the RNAi procedure, which is readily available and frequently used as an instrumental tool for the study of the transcriptional regulation mechanism upstream of the IAG-switch at early post-larval stages [30]. However, when studying the upstream sex-determining factors that affect the IAG-switch, the post-larval stage might prove to be too late for the extensive functional genomic efforts needed for such a challenging study in which some earlier interventions seem to be needed. Thus, a genomic editing platform based on CRISPR is required for manipulations at early embryonic stages. In recent years, the CRISPR-Cas genome-editing tool was developed [122] and revolutionized life science research, enabling the induction of targeted mutagenesis to determine the role of studied genes. In crustaceans, a few cases were described using CRISPR to determine the role of genes in different processes such as eye development [123,124], limb specification [125], chitinase activity [126], and more. Recently, Molcho et al. reported the first genomic editing platform in *M. rosenbergii*, presenting a CRISPR protocol through direct injection into one- to four-cell embryos, which results in entire organism genome editing [127]. In the prawn *E. carinicauda*, knocking out *EcIAG* and obtaining homozygous mutants with biallelic mutations led to sex reversal from males to neo-females, suggesting CRISPR/Cas9 genome editing as an effective tool for sex manipulation in crustaceans, further supporting monosex aquaculture [128]. However, the transformation of applied technologies from RNAi interventions that do not affect the genome into genetically modifying methods is debatable. Indeed, the first such modified organism has been approved for aquaculture [129], while regulators in several countries are debating whether minor CRISPR modifications will be more easily approved [130]. Given that the IAG-encoding gene was found to be residing on an autosome, gene editing can potentially be harnessed to manipulate the IAG-switch even in WW superfemales [42], which lack the male sex chromosome. Further research is therefore required to better understand the upstream regulation of *IAG* expression (comparing promoter methylation and acetylation patterns in males and females, and so forth).

## 6. Future Environmental Applications of the IAG Switch

Crustaceans’ hardiness and adaptability place them among the worst-known invasive species, subsequently threatening biodiversity [131] and ecosystem stability [132]. One of the leading invasive species on earth is the red swamp crayfish, *P. clarkii*, which is known to cause considerable environmental and economic damage [133]. Production of *P. clarkii* neo-females through IAG-switch manipulation, using RNAi-based biotechnology for all-male aquaculture similar to prawns [100], was suggested as a sustainable solution with the potential to greatly impact the aquaculture industry [134]. All-male crayfish aquaculture is expected to have higher incomes and is safer than a mixed population by preventing escapees from becoming invasive in regions where *P. clarkii* is non-native [134]. Savaya et al. (2020) also suggested a demographic model that explores the potential of stocking neo-females to control the invasive population in the wild by skewing the sex ratio of the population [134]. Utilizing additional control tools would be important for ensuring the efficacy of sex-skewing strategy in invasive population management [135]. To fulfill these potential solutions, further study is required on IAG-switch manipulations in *P. clarkii* to achieve fully functional sex reversal and monosex population production or the use of tools such as gene drive [136,137,138]. With respect to managing invasive species, the availability of *M.* rosenbergii monosex populations is not only advantageous for aquaculture mass production but also for better safeguarding the environment from aquaculture escapees in areas where *M. rosenbergii* could become an invasive species.

Another interesting environmental management strategy that was unlocked by the availability of monosex populations is the use of prawns as biological control agents. *M.* rosenbergii are voracious predators of pest freshwater snails. The possible use of non-breeding monosex prawn populations poses them as excellent biocontrol agents against such snails without the hazard of prawns establishing invasive populations that could harm the environment and its natural biodiversity. Among the pest snails most noted are freshwater snails that host the flatworm *Schistosoma* spp., which are responsible for the parasitic disease schistosomiasis (bilharzia). Schistosomiasis is a serious human health concern, mainly in Africa, with hundreds of millions of infected people [139]. Applying monosex biotechnology in areas with abundant snail hosts can break the disease cycle [140]. Apart from human health, snails are involved in the transmission of fish diseases in the aquaculture industry. Extensive damage is caused to freshwater aquaculture by parasitic diseases, such as those transmitted by the snails of the *Thiaridae* family, which hosts the disease-causing parasitic *Centrocestus* species [141]. The genus *Macrobrachium* is not vulnerable to Trematode infection [142,143], and studies demonstrated the high ability of the prawn to abolish snail hatchlings [140,144,145]. Savaya et al. (2020) demonstrated the first application of *M.* rosenbergii monosex populations as biocontrol agents in commercial aquaculture ponds. A field experiment in *Tilapia* aquaculture ponds in Israel included prawns as biocontrol agents, which significantly reduced the total biomass of *Melanoides tuberculate* and *Thiara scabra* snails of all size classes, and the number of parasites per fish was lower in ponds with prawns than in the control (no-prawns) ponds [146]. Another invasive snail with a high impact on global agriculture is the invasive freshwater apple snail (*Pomacea* spp.), one of the major unresolved problems worldwide with a significant impact on global rice production, affecting not only economics but also natural ecosystems and potentially also causing health issues [147]. Utilizing monosex prawn populations has been suggested as a sustainable solution against the invasive apple snails and presents an effective biocontrol method under different conditions. Future field validation experiments are required for environmental risk assessment [144,148]. With respect to the treatment of pest snails in agriculture, aquaculture, and even human disease, prawns as biocontrol agents could become a crop by themselves, contributing additional quality protein sources or extra income, thus suggesting a win-win solution and providing a sustainable solution that has the cultural, economic, and environmental incentives to be propagated in perpetuity.

## Figures and Tables

**Figure 1 ijms-24-17433-f001:**
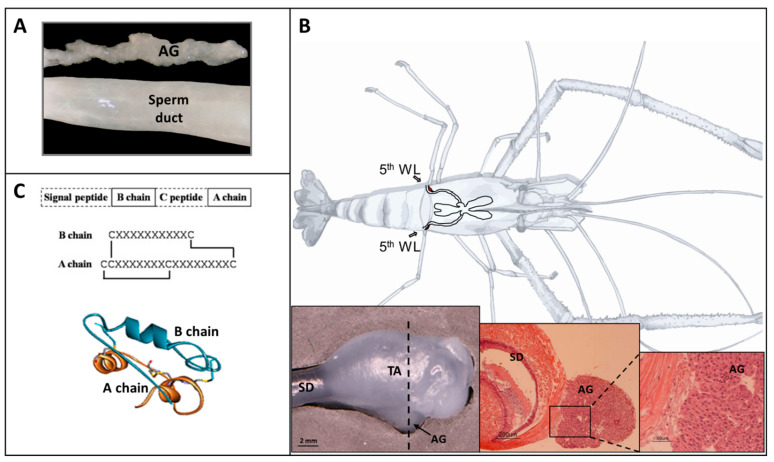
Monosex population aquaculture of the giant freshwater prawn *Macrobrachium rosenbergii*: ‘the tools of the trade’. (**A**) The androgenic gland (AG) was first discovered in a decapod in 1947, although it was not until 2007 that the first decapod AG-derived insulin-like peptide (IAG) encoding sequence (*IAG*) was identified. This photo shows the AG situated alongside the sperm duct in the Australian redclaw crayfish, *Cherax quadricarinatus*. (**B**) Proximal parts of the male reproductive systems of *M. rosenbergii* males; cross-section of the terminal ampullae (TA) with the sperm duct (SD) and AG area. The cells of this gland, following an endocrine manipulation to induce its hypertrophy, are dispersed and injected into females to induce sex change into neo-males, which generates all-female populations. Drawing—Dr. Shaul Raviv (**C**) Following decades of research prior to the advent of high-throughput sequencing, the IAG is now identified in multiple decapod species, where it is found to primarily express in the AG. The gene encodes a signal peptide, followed by a B chain, a C peptide, and an A chain. During translation, the signal peptide is cleaved off, followed by cleavage of the C peptide, leaving the B chain and A chain interlinked with two disulphide bonds and another disulphide bond within the A chain (linear model), rendering the mature IAG with an insulin-like three-dimensional structure signature. Silencing *IAG* in early stage of male development enables sex change into neo-females, which generates all-male populations.

**Figure 2 ijms-24-17433-f002:**
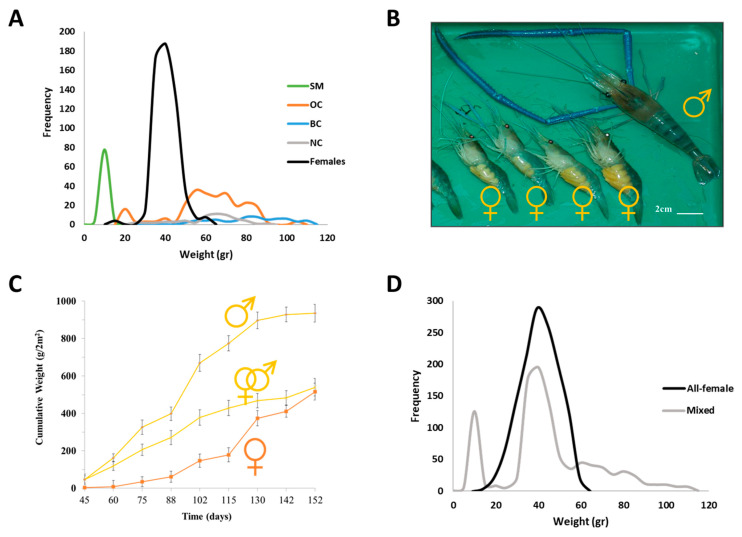
Monosex population aquaculture of the giant freshwater prawn *Macrobrachium rosenbergii*: both all-male and all-female populations might be advantageous. (**A**) Weight distribution of male morphotypes and females in a mixed population. Sample groups of females, small male (SM), orange-claw male (OC), blue-claw male (BC), and no-claw male (NC) under extensive stocking density. (**B**) A large dominant male harvested with four females from the same population. Note that all the females carry eggs and are of uniform size. Photograph—Tomer Ventura. (**C**) A small-scale pond study concluded that all-male giant freshwater prawn populations, when selectively harvested, produce double the cumulative yield when compared with mixed-sex and all-female populations. (**D**) Weight distribution of all-female and mixed population at the end of the grow-out season. Females grow uniformly, with better survival rates, even at high stocking densities.

**Figure 3 ijms-24-17433-f003:**
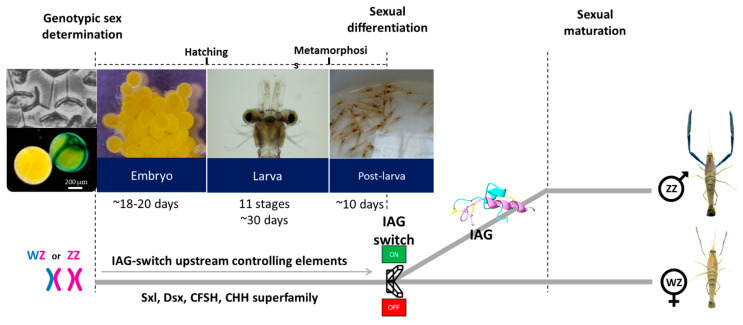
From fertilization to maturation in the giant freshwater prawn *Macrobrachium rosenbergii* through the IAG-switch. The process starts with the genotype determination, WZ or ZZ, followed by developmental stages of embryos, larvae, and early post-larvae. An individual with a ZZ set of chromosomes develops an androgenic gland (AG), which secretes the IAG, leading to the development of a functional, mature male phenotype. In an individual with the WZ set of chromosomes, the AG is absent, so there is no IAG secretion, and a functional, mature female phenotype is exhibited. This system could be manipulated as a sex-differentiating switch, thus termed the “IAG-switch”. However, the controlling elements upstream the IAG-switch are not yet elucidated.

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
