# Peer review of "Monosex Populations of the Giant Freshwater Prawn Macrobrachium rosenbergii—From a Pre-Molecular Start to the Next Generation Era"

_ijms, 2023, doi:10.3390/ijms242417433_

Round 1

Reviewer 1 Report

Comments and Suggestions for Authors

This review is of great significance as it summarizes basic research and its application for the mechanism of sex differentiation in decapod crustaceans promoted by the authors' group.

1)    However, despite the title "M. rosenbergii", the content covers a wide range of marine as well as freshwater species. The title should be reconsidered.

2)    Vertebrate sex-steroids (and the related substances) are found several decapods. But, it is very simply described that they are not likely to play the significant role in the sex differentiation of decapods in the first part of the MS. A little more detailed explanation will be required.

3)    Gender differences in behavior would also be a consideration when conducting a monosex culture of decapods. In this regard, it is advisable that the related discussion should be added based on some reports such as the important one by the authors (https://doi.org/10.1242/jeb.00335), a report on feeding (https://doi.org/10.1016/j.applanim.2020.104946), and so on.

L63: “similar”

L415: “Journal of Fisheries of China” may be better.

L493: The article ID should be presented.

L499: “Chung J.S.”

Author Response

Reviewer #1 report and responses:

This review is of great significance as it summarizes basic research and its application for the mechanism of sex differentiation in decapod crustaceans promoted by the authors' group.

  • However, despite the title " rosenbergii", the content covers a wide range of marine as well as freshwater species. The title should be reconsidered.

Response: we thank the reviewer for acknowledging the significance of this review. Since monosex was achieved to date only in M. rosenbergii, we focus on this species in the title and throughout the text and provide examples of different reproductive strategies and the knowledge gained towards broader impact across species, but the focus still remains largely on M. rosenbergii monosex.

  • Vertebrate sex-steroids (and the related substances) are found several decapods. But, it is very simply described that they are not likely to play the significant role in the sex differentiation of decapods in the first part of the MS.A little more detailed explanation will be required.

Response: We have added the following in lines 22-29: “Studies have investigated the presence of sex steroids and their potential role in sexual development in crustaceans; Sex steroids, including androgen, estrogen, and progestogen, have been identified in decapod tissues, such as the gonads and eyestalk (Lin et al., 2019; Okumura & Sakiyama, 2004; Wang et al., 2022; T. Wang et al., 2023). Although some studies have explored the effects of hormone treatments on sexual development and reproductive processes (M. Liu et al., 2018; Sugestya et al., 2018; Yano, 1985; Yano, 1987), the literature regarding the underlying mechanisms and factors influencing the outcomes remains limited and lacks clear evidence.”

  • Gender differences in behavior would also be a consideration when conducting a monosex culture of decapods. In this regard, it is advisable that the related discussion should be added based on some reports such as the important one by the authors (https://doi.org/10.1242/jeb.00335), a report on feeding (https://doi.org/10.1016/j.applanim.2020.104946), and so on.

Response: We have added the following in lines 145-152 : M. rosenbergii presents clear differences between sexes. Males display morphotypic differentiation while females grow more uniformly, making monosex a clear advantage. The AG is known to be key for this change in behavior, as was exemplified in C. quadricarinatus where AG implanted into females induces male-like behaviors (Barki et al., 2003). While sex-specific behaviors remain largely un-explored across farmed decapods, evidence suggests monosex would be advantageous across decapods, as for example in the case of the commercially most important marine shrimp Litopenaeus vannamei where subtle sex-biased behaviors were recorded (Bardera et al., 2020).

The following comments were all changed as advised:

L63: “similar”

L415: “Journal of Fisheries of China” may be better.

L493: The article ID should be presented.

L499: “Chung J.S.”

Reviewer 2 Report

Comments and Suggestions for Authors

Review for the paper “Monosex populations of the giant freshwater prawn Macrobrachium rosenbergii – from a pre-molecular start to the next generation era” by Melody Wahl, Tom Levy, Tomer Ventura and Amir Sagi submitted to "International Journal of Molecular Sciences".

General comment.

The authors described brief history of the discoveries of the AG and the IAG in crustaceans, studies focused on manipulations with AG levels using surgical methods and RNA interference gene knockdown, controlling elements of the IAG-Switch, molecular and genomic implications, Distinct biotechnologies for monosex populations of prawns, future environmental applications of the IAG switch.

These data are closely related to information recently summarized in the following review paper

Levy, T. and Sagi, A. (2020) The “IAG-Switch” – A Key Controlling Element in Decapod Crustacean Sex Differentiation. Front. Endocrinol. 11:651. doi: 10.3389/fendo.2020.00651

https://www.frontiersin.org/articles/10.3389/fendo.2020.00651/full

In general, only sections 4 and 6 contain new information, but section 4 is far from the topic of the review.

I think that the novelty of this review is very low.

Some additional comments are given below.

Major concerns.

Abstract. The authors should expand the abstract with the most important findings in the field and indicate the most important implications for aquaculture.

Surprisingly, this review does not contain a classical introduction. The paper starts with a historical sketch and does not contain any data about the object of study (shrimp), its aquaculture, main directions and problems, relevance of this topic and finally the aim of this study is not presented.

1. Section 1. This section requires relevant citations (L 31-34).

Section 1 contains only one subsection, which raises the question whether this division is necessary. The same applies to section 2.

Logically, Section 4 should follow Section 5.

An appropriate conclusion is needed.

Figure captions. The authors should follow MPDI style when citing references.

The authors should consider the following recent publications.

Xu, H. J., et al. (2022). Full functional sex reversal achieved through silencing of mrodmrt11e gene in Macrobrachium rosenbergii: production of all-male monosex freshwater prawn. Frontiers in Endocrinology, 12, 1867.

Qian, H., et al. (2022). Transcriptome analysis of the post-larvae of giant freshwater prawn (Macrobrachium rosenbergii) after IAG gene knockdown with microRNA interference. General and Comparative Endocrinology, 325, 114054.

Sun, R., Li, Y. (2021). A sexreversing factor: insulinlike androgenic gland hormone in decapods. Reviews in Aquaculture, 13(3), 1352-1366.

Specific remarks.

L 35. Consider replacing " malacostraca" with " Malacostraca".

L 55. Consider replacing " with regards" with " with regard".

L 56. Consider replacing " In  Isopods" with " In  isopods".

L 63. Consider replacing " Later, Similar" with " Later, similar".

L 66. Consider replacing " suggested steroids to act" with " suggested that steroids act".

L 84. Consider replacing " crustacean" with " crustaceans".

L 99. Consider replacing " in contrary" with " in contrast".

L 113. Consider replacing " as a replacement of" with " as a replacement for".

L 118. Consider replacing " where  the  use" with " when  the  use".

L 127. Consider replacing " Although M. rosenbergii some males grow larger than females and reaching the highest sizes in a mix population" with " Although in M. rosenbergii, some males grow larger than females and reach the highest sizes in a mixed population".

L 136. Consider replacing " depend" with " depends".

L 152. Consider replacing " upstream the IAG-switch" with " upstream of the IAG-switch".

L 160. Consider replacing " which resulted with" with " which resulted in".

L 170. Consider replacing " crucial role at the early" with " crucial role in the early".

L 182. Consider replacing " ablation in males cause" with " ablation in males causes".

L 214. Consider replacing " resulted in significant decrease of" with " resulted in a significant decrease in".

L 216. Consider replacing " candidates upstream the" with " candidates upstream of the".

L 224. Consider replacing " in several crustacean" with " to several crustacean".

L 248. Consider replacing " debateable. Indeed, first" with " debatable. Indeed, the first".

L 285. Consider replacing " need of manual segregation" with " need for manual segregation".

L 296. Consider replacing " performances of the WW all-female population was comparable" with " performances of the WW all-female population were comparable".

L 320. Consider replacing " pose  them" with " poses  them".

L 329. Consider replacing " Macrobrachium genus are not vulnerable to Trematode" with " The genus Macrobrachium is not vulnerable to trematode".

L 336. Consider replacing " high impact on the global agriculture" with "a high impact on global agriculture".

Comments on the Quality of English Language

The English is good.

Author Response

Reviewer #2 report and responses:

The authors described brief history of the discoveries of the AG and the IAG in crustaceans, studies focused on manipulations with AG levels using surgical methods and RNA interference gene knockdown, controlling elements of the IAG-Switch, molecular and genomic implications, Distinct biotechnologies for monosex populations of prawns, future environmental applications of the IAG switch.

These data are closely related to information recently summarized in the following review paper

Levy, T. and Sagi, A. (2020) The “IAG-Switch” – A Key Controlling Element in Decapod Crustacean Sex Differentiation. Front. Endocrinol. 11:651. doi: 10.3389/fendo.2020.00651

https://www.frontiersin.org/articles/10.3389/fendo.2020.00651/full

In general, only sections 4 and 6 contain new information, but section 4 is far from the topic of the review.

I think that the novelty of this review is very low.

Response: According to the reviewer's comment, we have added the following to increase the novelty of the review. Section 3 is now includes recent studies regarding the controlling elements upstream the IAG-switch; to expand its novelty we have also added Table 1 which summarized the recent functional studies on genes suggested as sexual development regulation upstream the IAG-Switch, and the following in lines 214-216: “The research on the upstream controlling elements of sexual differentiation through the IAG-switch is in progress, with functional studies suggesting diverse candidates for this role (Table 1).” In addition, we have added the following in lines 22-29: “Studies have investigated the presence of sex steroids and their potential role in sexual development in crustaceans; Sex steroids, including androgen, estrogen, and progestogen, have been identified in decapod tissues, such as the gonads and eyestalk (Lin et al., 2019; Okumura & Sakiyama, 2004; Wang et al., 2022; T. Wang et al., 2023). Although some studies have explored the effects of hormone treatments on sexual development and reproductive processes (M. Liu et al., 2018; Sugestya et al., 2018; Yano, 1985; Yano, 1987), the literature regarding the underlying mechanisms and factors influencing the outcomes remains limited and lacks clear evidence.”

Some additional comments are given below.

Major concerns.

Abstract. The authors should expand the abstract with the most important findings in the field and indicate the most important implications for aquaculture.

Response:  As agreed by Reviewer #1, this review is “of great significance”. It is extending on what was reviewed in the former review by adding new knowledge and train of thought, including the reference to gene editing and how we speculate this will impact the research and development field and the environmental implications of monosex. 

Surprisingly, this review does not contain a classical introduction. The paper starts with a historical sketch and does not contain any data about the object of study (shrimp), its aquaculture, main directions and problems, relevance of this topic and finally the aim of this study is not presented.

Response: this review does not aim at covering aquaculture of M. rosenbergii and key challenges, but rather focus on the molecular aspects of monosex. As such, the first section provides a brief history of monosex development, presenting the AG and IAG which are the key known components. This brief historical perspective serves as the introduction on which this review is established.

Section 1. This section requires relevant citations (L 31-34).

Response: the following reference was added: https://doi.org/10.1242/jeb.047183

Section 1 contains only one subsection, which raises the question whether this division is necessary. The same applies to section 2.

Response: the sectioning clearly identifies the literature covered in each section and is intended to help guide the readers that are more knowledgeable to avoid reading through literature they are already aware of.

Logically, Section 4 should follow Section 5.

Response: we replaced sections 4 and 5.

An appropriate conclusion is needed.

Response:

Figure captions. The authors should follow MPDI style when citing references.

 Response:

The authors should consider the following recent publications.

Xu, H. J., et al. (2022). Full functional sex reversal achieved through silencing of mrodmrt11e gene in Macrobrachium rosenbergii: production of all-male monosex freshwater prawn. Frontiers in Endocrinology, 12, 1867.

Qian, H., et al. (2022). Transcriptome analysis of the post-larvae of giant freshwater prawn (Macrobrachium rosenbergii) after IAG gene knockdown with microRNA interference. General and Comparative Endocrinology, 325, 114054.

Sun, R., Li, Y. (2021). A sex‐reversing factor: insulin‐like androgenic gland hormone in decapods. Reviews in Aquaculture, 13(3), 1352-1366.

 Response: The following reference was added in Table 1: https://doi.org/10.3389/fendo.2021.772498

Specific remarks.

All minor comments below were addressed and the text changed as suggested.

L 35. Consider replacing " malacostraca" with " Malacostraca".

L 55. Consider replacing " with regards" with " with regard".

L 56. Consider replacing " In  Isopods" with " In  isopods".

L 63. Consider replacing " Later, Similar" with " Later, similar".

L 66. Consider replacing " suggested steroids to act" with " suggested that steroids act".

L 84. Consider replacing " crustacean" with " crustaceans".

L 99. Consider replacing " in contrary" with " in contrast".

L 113. Consider replacing " as a replacement of" with " as a replacement for".

L 118. Consider replacing " where  the  use" with " when  the  use".

L 127. Consider replacing " Although M. rosenbergii some males grow larger than females and reaching the highest sizes in a mix population" with " Although in M. rosenbergii, some males grow larger than females and reach the highest sizes in a mixed population".

L 136. Consider replacing " depend" with " depends".

L 152. Consider replacing " upstream the IAG-switch" with " upstream of the IAG-switch".

L 160. Consider replacing " which resulted with" with " which resulted in".

L 170. Consider replacing " crucial role at the early" with " crucial role in the early".

L 182. Consider replacing " ablation in males cause" with " ablation in males causes".

L 214. Consider replacing " resulted in significant decrease of" with " resulted in a significant decrease in".

L 216. Consider replacing " candidates upstream the" with " candidates upstream of the".

L 224. Consider replacing " in several crustacean" with " to several crustacean".

L 248. Consider replacing " debateable. Indeed, first" with " debatable. Indeed, the first".

L 285. Consider replacing " need of manual segregation" with " need for manual segregation".

L 296. Consider replacing " performances of the WW all-female population was comparable" with " performances of the WW all-female population were comparable".

L 320. Consider replacing " pose  them" with " poses  them".

L 329. Consider replacing " Macrobrachium genus are not vulnerable to Trematode" with " The genus Macrobrachium is not vulnerable to trematode".

L 336. Consider replacing " high impact on the global agriculture" with "a high impact on global agriculture".

Reviewer 3 Report

Comments and Suggestions for Authors

In this review, the authors have highlighted the pivotal roles of insulin-like peptides (ILPs), including the insulin-like AG hormone (IAG), in regulating various aspects of development, growth, and metabolism across a spectrum of organisms, encompassing humans and other animals. The focus of this review centers on elucidating the effects of the insulin-like AG hormone (IAG) in the context of sexual manipulation.

IAG exhibits a striking structural similarity to other insulin-like peptides observed in specific species. The authors have presented the structure of IAG, which bears resemblance to insulin-like peptides. Conducting an alignment of IAG with these related insulin-like peptides will enable a comparative examination of their sequence characteristics. The integration of these alignment results into the manuscript would not only enhance our comprehension of the subject matter but also offer valuable insights into the study of insulin-like peptides in other species.

To enhance the clarity of the intricate sexual development pathways among decapods, please add more details on the regulatory pathway in Figure 3 using the results from a well-deciphered species. Such an illustrative representation will not only promote a comprehensive understanding of IAG in other species but also serve as a valuable visual aid in elucidating this complex process. 

Author Response

We have added a supplementary file which addresses the IAG in context of other insulin-like peptides. Please note that this phylogenetic analysis was already previously reported elsewhere (Veenstra J., 2020; https://peerj.com/articles/9534/), we trust that the required information, with the citation of this resource addresses the reviewer comment regarding phylogenetics.

Round 2

Reviewer 2 Report

Comments and Suggestions for Authors

Second review for the paper “Monosex populations of the giant freshwater prawn Macrobrachium rosenbergii – from a pre-molecular start to the next generation era” by Melody Wahl, Tom Levy, Tomer Ventura and Amir Sagi submitted to "International Journal of Molecular Sciences".

The currently revised manuscript does not adequately address the fundamental issues that were previously raised:

1. The abstract remained deficient in encapsulating the critical aspects of the study. Essential details that would provide a comprehensive understanding were conspicuously missing.

2. The introduction, in my view, did not meet the standard expectations. This section should be particularly devoted to outlining the objective and the distinct novelty of the research conducted, as per the conventional structure of a scientific paper. A historical overview, though informative, is not synonymous with the introductory part of a scholarly article.

3. Another glaring omission was the absence of a conclusion section. An academic paper necessitates the inclusion of this part to summarize and finalize the key points effectively.

4. In the revised manuscript, there appears to be a disconnect: all figures cited in the text were omitted from the submission, creating an ambiguity in understanding the reported findings.

5. The references and citations do not conform to the formatting protocol of the MDPI.

6. The inclusion of a new table in the recent revision does not significantly differentiate the paper from the prior review published in FMS, implying potential issues of overlap and a lack of novelty. It is essential that the content is distinctively updated to reflect the unique contribution of the current study.

Given these persistent concerns, I am compelled to maintain my initial recommendation of rejection for this manuscript.

Comments on the Quality of English Language

Minor

Author Response

We have added supplementary data, which includes a list of IAG, ILPs, and representative members of the insulin superfamily protein sequences, and highlighted the insulin structure (S1). In addition, we have conducted an alignment of the above sequences' conserved characteristics (S2), as suggested by the reviewer. The following (underlined) was added in lines 74-78: “It was not until 2007 when the first AG-specific gene, expressed exclusively in males, was discovered in the crayfish Cherax quadricarinatus, termed insulin-like AG factor (IAG) (Manor et al., 2007) (Figure 1C), followed by its discovery in M. rosenbergii (Ventura et al., 2009), demonstrating structural similarity to the insulin superfamily (Supplementary data S1, S2).”

To enhance the clarity of the intricate sexual development pathways among decapods, please add more details on the regulatory pathway in Figure 3 using the results from a well-deciphered species. Such an illustrative representation will not only promote a comprehensive understanding of IAG in other species but also serve as a valuable visual aid in elucidating this complex process.